# Ex Vivo Study of Colon Health, Contractility and Innervation in Male and Female Rats after Regular Exposure to Instant Cascara Beverage

**DOI:** 10.3390/foods13162474

**Published:** 2024-08-06

**Authors:** Paula Gallego-Barceló, David Benítez-Álvarez, Ana Bagues, Blanca Silván-Ros, Alba Montalbán-Rodríguez, Laura López-Gómez, Gema Vera, María Dolores del Castillo, José A. Uranga, Raquel Abalo

**Affiliations:** 1Department of Basic Health Sciences, Faculty of Health Sciences, University Rey Juan Carlos (URJC), 28922 Alcorcón, Spain; paula.gallego@urjc.es (P.G.-B.); david.beniteza@urjc.es (D.B.-Á.); blanca.silvan@urjc.es (B.S.-R.); alba.montalban@urjc.es (A.M.-R.); laura.lopez.gomez@urjc.es (L.L.-G.); gema.vera@urjc.es (G.V.); jose.uranga@urjc.es (J.A.U.); 2High Performance Research Group in Physiopathology and Pharmacology of the Digestive System (NeuGut-URJC), University Rey Juan Carlos, 28922 Alcorcón, Spain; mdolores.delcastillo@csic.es (M.D.d.C.);; 3Institute of Human Genetics, Faculty of Medicine and University Hospital Cologne, University of Cologne, Kerpener Street 34, 50931 Cologne, Germany; 4Center for Molecular Medicine Cologne (CMMC), Faculty of Medicine and University Hospital Cologne, University of Cologne, Robert-Koch-Street 21, 50931 Cologne, Germany; 5High Performance Research Group in Experimental Pharmacology (PHARMAKOM-URJC), University Rey Juan Carlos, 28922 Alcorcón, Spain; 6Associated I+D+i Unit to the Institute of Medicinal Chemistry (IQM), Scientific Research Superior Council (CSIC), 28006 Madrid, Spain; 7Department of Biochemistry, Medical University of Lodz, 92-215 Lodz, Poland; 8Food Bioscience Group, Department of Bioactivity and Food Analysis, Institute of Food Science Research (CIAL) (CSIC-UAM), Nicolás Cabrera Street, 9, 28049 Madrid, Spain; 9Working Group of Basic Sciences on Pain and Analgesia of the Spanish Pain Society, 28046 Madrid, Spain; 10Working Group of Basic Sciences on Cannabinoids of the Spanish Pain Society, 28046 Madrid, Spain

**Keywords:** antioxidants, coffee by-products, colon contractility, Instant Cascara, muscarinic receptors, myenteric plexus, organ bath, rat, sex, substance P

## Abstract

Instant Cascara (IC) is a sustainable beverage made from dried coffee cherry pulp, a by-product of coffee processing. It is rich in nutrients and bioactive compounds and has a high concentration of antioxidants. This study explored the impact of regular IC consumption on colonic motor function and innervation. Over a period of 4 weeks, male and female healthy rats were given drinking water containing 10 mg/mL of IC. Thereafter, colon samples were obtained to evaluate the longitudinal (LM) and circular (CM) smooth muscle contractile response to acetylcholine (ACh) and electrical field stimulation (EFS) in an organ bath, before and after atropine administration (10^−6^ M). Histological and immunohistochemical analyses assessed colon damage, muscle thickness, and immunoreactivity to substance P (SP) and neuronal nitric oxide synthase (nNOS). ACh and EFS induced similar responses across groups, but the CM response to EFS was greater in females compared with males, despite their lower body weight. Atropine completely blocked the response to ACh but only partially antagonized the neural response to EFS, particularly that of CM in females treated with IC, which had a greater liquid intake than those exposed to water. However, in the myenteric ganglia, no statistically significant differences were observed in SP or nNOS. Our results suggest that regular IC exposure may enhance specific neural pathway functions, particularly in females, possibly due to their increased IC consumption.

## 1. Introduction

The gastrointestinal system is vital for human health, converting food into essential nutrients and eliminating waste products. The colon, a key part of this system, absorbs water and forms feces, helping to maintain the body’s fluid balance [1]. It also houses the gut microbiome, which influences our metabolic, immune, and neurological functions [2]. The efficient functioning of the colon is crucial for overall health, demonstrating the importance of the digestive system in our well-being. Additionally, any disruption in its function can lead to significant health issues, such as constipation or diarrhea, decreasing quality of life [1]. Maintaining a balanced diet and staying well hydrated are essential for a healthy colon and for ensuring the digestive system functions effectively [1].

The gut wall, including that of the colon, is intrinsically innervated by the enteric nervous system (ENS). The ENS comprises an intricate network of enteric neurons that interact with enteric glial cells (EGCs) and interstitial cells of Cajal (ICCs) [3,4]. The ENS can obtain information, process it and induce a response on its own. In this way, it can coordinate the functions of the gastrointestinal tract independently of the central nervous system (CNS) [3,4,5,6]. The ENS and CNS communicate through the afferent and efferent pathways of the autonomic nervous system (ANS), forming the gut–brain axis, in which the gut microbiota also participates [7,8,9].

Neurons of the ENS are grouped into two plexuses along the gastrointestinal tract: the submucosal or Meissner’s plexus and the myenteric or Auerbach’s plexus. The myenteric plexus is situated between the layers of circular and longitudinal smooth muscle, and it intrinsically regulates motor function [4,10]. In the myenteric plexus, the main excitatory neurotransmitter is acetylcholine (ACh), a muscarinic receptor agonist; other excitatory neurotransmitters are tachykinins (neurokinin A and substance P (SP)) and serotonin (5-HT). Additionally, inhibitory motor neurons can release various neurotransmitters for muscle relaxation such as nitric oxide (NO), ATP, gamma-aminobutyric acid (GABA) or catecholamines amongst others [4,10,11].

An alteration in the balance of excitatory and inhibitory neurotransmitters in the ENS, in their modulation by the CNS, the endocrine system (hormones) and the microbiota, or in the muscle or ICC function, can lead to serious gastrointestinal motor dysfunctions (i.e., constipation and/or diarrhea), such as those related to irritable bowel syndrome (IBS), diabetes and Parkinson’s disease [8,9,10,12].

Instant Cascara (IC) is a beverage produced from the dried pulp of ripe coffee cherries, a relevant coffee by-product, and it includes bioactive ingredients such as caffeine and phenolic compounds [13]. Both caffeine and phenolic compounds are well known for their antioxidant and anti-inflammatory properties [14,15]. Interestingly, they may have other effects on the gastrointestinal tract. For example, caffeine specifically acts on the colon, invigorating the propulsive motility patterns essential for consistent bowel movements. Its effect, however, is not limited to the muscle; it also modulates both the internal and external neural pathways which orchestrate gut motility [16]. Although the phenolic compounds in IC have not been shown to have a direct effect on gastrointestinal motility, they may support the neural regulation of the gut by potentially reducing or preventing oxidative stress in neural tissues [14,15]. Together, these bioactive elements of IC may work in concert to foster an environment conducive to a healthy colonic motility. 

The colonic physiological effects resulting from regular IC consumption could stem from the bioactive compounds themselves or their metabolites, interacting with nerve terminals in the intestinal lumen or following systemic absorption [17,18,19]. Additionally, considering IC as a complex food matrix, there is the potential for antagonistic or synergistic interactions among compounds, altering their ultimate influence on the gastrointestinal system [18]. Moreover, these changes might be influenced by individual factors, like biological sex. However, it is remarkable that previous research has often focused on the local effects of drinks or their components when applied directly to isolated gastrointestinal tissues and have rarely considered sex as an influential factor [20]. Importantly, studies on the specific effects of regular consumption of foods and drinks on the gut wall and its innervation cannot be easily performed in healthy people, highlighting the importance of animal studies, which remain essential for understanding the intricate interactions and physiological effects of dietary interventions. 

Whereas in a previous *in vivo* study in healthy male and female rats, we did not observe any significant difference in gastrointestinal transit due to sex or IC consumption [21], the specific effects of the regular intake of IC on the functionality of the colonic muscle, its intrinsic innervation or both have not been studied in either sex. These effects might be relatively permanent and play an adaptive role to those occurring *in vivo* or be silent under control *in vivo* conditions but may underlie improved or worsened functions under abnormal situations.

Therefore, this study aimed to investigate ex vivo the effects of regular IC exposure on colonic contractility and innervation in healthy male and female animals.

## 2. Resources and Procedures

### 2.1. Ethical Declaration

The research protocol received authorization from the Ethics Committee of Rey Juan Carlos University (URJC) and Comunidad de Madrid (PROEX-059/2018). It adhered to the European Community Council Directive of 22 September 2010 (2010/63/EU) and Spanish legislation (Law 32/2007, RD 53/2013, and Order ECC/566/2015) regarding the protection of animals used in scientific research. The protocol was designed to minimize animal suffering and distress and aimed to reduce the number of animals involved.

### 2.2. Instant Cascara Beverage

Dried ripe coffee cherry pulp (fruit cascara) from the Arabica species and Tabi variety, supplied by SUPRACAFÉ S.A. (Móstoles, Madrid, Spain), was used to produce the soluble powder known as IC. The coffee fruit cascara underwent a wet method involving pulping, sun drying for a week to reduce humidity to 10% and a sanitation process using irradiation by IONISOS Ibérica. This product was submitted to aqueous extraction using 50 g/L at 100 °C for 10 min, resulting in a 20% yield. The extract was then filtered through a 250 μm sieve and freeze-dried to produce the IC powder. This method is described in the patent WO2013004873A [22]. Finally, the IC beverage was prepared by dissolving 10 mg of IC powder per mL of filtered sterile water at room temperature. The concentration of IC was chosen based on our previously published studies [14,21]. This concentration was well accepted by both male and female rats [21]. As such, the IC beverage has the composition shown in Table 1.

### 2.3. Experimental Groups and Study Animals 

Young adult Wistar rats (2–3 months old) were involved in this study, including males (250–350 g, *n* = 6) and females (200–250 g, *n* = 6), obtained from the URJC Veterinary Unit. The rats were housed in standard transparent cages measuring 60 cm × 40 cm × 20 cm, separated by sex, with 3 rats per cage. They had free access to chow pellets (SAFE D40 diet) and sterile tap water. Environmental conditions were controlled, maintaining a temperature of 20 °C and humidity at 60%, with a 12 h light/dark cycle (lights on from 8:00 to 20:00). Rats were randomly allocated into four experimental groups according to their sex and the type of beverage they received. Half of the male and female rats were given tap water in their drinking bottles, while the remaining rats were provided with the IC beverage.

### 2.4. Methodological Approach 

Figure 1 provides a summary of the experimental protocol. Animals were given either water or IC beverage from week 1 until sacrifice (week 4). During weeks 1–3, body weight, as well as food and liquid consumption, were monitored. After sacrifice (week 4), the colon was obtained for organ bath studies and histology/immunohistochemistry.

Prior to sacrifice, vaginal cytology was conducted on female rats to determine the estrous cycle phase. Vaginal smears were collected using a cotton-tipped applicator inserted 2 cm into the vaginal opening, rotated three times, and then spread onto slides for hematoxylin and eosin staining. Microscopic observation allowed the determination of the estrous cycle phase: proestrous, estrous, metestrous, or diestrous [24,25].

Experienced researchers performed all analyses without knowing which treatments the animals received.

### 2.5. Colon Wall Health: Histological Analysis

Distal colon samples were fixed in buffered 4% formaldehyde and embedded in paraffin. Subsequently, 5 μm sections were stained with hematoxylin and eosin (HE) or toluidine blue and studied under a Zeiss Axioskop 2 microscope (Zeiss, Aalen, Germany) equipped with the image analysis software package LAS X 3.7 (Leica, Wetzlar, Germany). 

The analysis of colonic damage was made in ten random fields per measured section under a 40× objective with a total of three independent sections of the colon tissue per animal. Histological damage score for the colon was assessed by criteria from Ranganathan et al. [26] based on the severity of the inflammation (0–3: none, slight, moderate, severe); depth of the injury (0–3: none, mucosal, mucosal and submucosal, transmural); crypt damage (0–4: none, basal 1/3 damaged, basal 2/3 damaged, only surface epithelium intact, entire crypt, and epithelium lost); and percentage of the involved area (0–4: 0, 1–10, 10–25, 25–50, and 50–100%). All scores on the individual parameters together result in a total score ranging from 0 to 14. In addition, the width of circular and longitudinal muscle layers was measured [27]. Immunohistochemistry was also performed on colonic samples, as described below (see Section 2.7. Innervation).

### 2.6. Colon Contractility: Organ Bath Study

To determine the effect of IC intake specifically on colonic contractility, organ bath experiments were performed as previously described [28]. The colon was placed on a Petri dish lined with Sylgard^®^ and filled with Krebs solution. The fat and mesentery were removed, and a longitudinal cut was made along the mesenteric border, so that the colonic lumen was opened, and the preparation was stretched, leaving the mucosal layer exposed. Subsequently, the mucosa and submucosa were removed by sharp dissection. Both circular muscle (CM) and longitudinal muscle (LM) strips were obtained by cutting perpendicular or parallel to the longitudinal axis of the colon, respectively. Therefore, 4 strips of CM and 4 of LM were obtained, with an approximate size of 10 × 5 mm each. A schematic representation can be observed in Figure 2.

The strips were placed in organ bath cups containing 10 mL of Krebs solution, maintained at 37 °C and tensioned to a load of 1 g, with carbogen (95% O_2_–5% CO_2_) aerating the solution. Each strip was secured at one end with a thread to a lower electrode and passed through a second, ring-shaped electrode (electrode separation: 0.7–1 cm). The thread was then attached to an isometric force transducer, allowing the measurement of strip contractile activity. Data were visualized, recorded, and analyzed using the Labchart program (version 8.1, ADInstruments Ltd., Oxford, UK) and an amplifier system (PowerLab/4e, ADInstruments Ltd., Oxford, UK).

The organ bath experiment procedure is depicted in Figure 3. Following a stabilization period of 60 min, during which Krebs solution was refreshed every 20 min and tension adjustments were made to maintain initial conditions, when necessary, strip functionality was assessed by adding potassium chloride (KCl) at 50 mM. Subsequently, two Krebs solution renewals were conducted over a 10 min interval, with the first renewal occurring 3 min after KCl administration.

The experimentation encompassed testing two distinct stimuli: electrical and chemical (ACh) [28,29]. Initially, electrical field stimulation (EFS) was applied, delivering trains of electrical pulses lasting 10 s each (with a pulse duration of 0.3 ms) at a voltage of 100 V, with frequencies increasing from 0.1 to 20 Hz every 5 min. Following a five-minute interval after the final EFS, the contractile response to ACh (a type 2 and 3 muscarinic receptor agonist) at different concentrations (ranging from 10^−8^ to 10^−5^ M) was assessed. Each ACh concentration was administered every 10 min, with two replacements of Krebs solution in between. The first solution renewal occurred immediately after reaching the maximum contraction response and plateau, typically around 3 min post drug administration.

Following the completion of both curves, atropine, a non-selective muscarinic antagonist, was added to the organ bath at a concentration of 10^−6^ M. Subsequently, the EFS curve was repeated, mirroring the initial phase of the protocol. Following this, the highest concentration of ACh (10^−5^ M) was used. Krebs solution was not replaced before any stimulus in this instance.

After the administration of ACh, Krebs solution was refreshed twice within the subsequent 10 min, and 50 mM of KCl was added to examine any potential impacts of the experimental protocol on the preparations. 

The data collected from the organ bath experiments underwent analysis using LabChart software. For the contractile response induced by KCl and ACh, the amplitude of the phasic (PA) and tonic (TA) components of the contraction was quantified (Figure 4). Preparations exhibiting PA values below 0.1 g for LM and 0.4 g for CM in response to the initial KCl stimulation, as well as those that ruptured during the experiment, were excluded from the analysis.

Regarding EFS, both the maximum amplitude of the response observed during the 10 s stimulation (Amplitude 1, A_1_) and the maximum response following the cessation of stimulation (Amplitude 2, A_2_) were measured (Figure 4).

### 2.7. Colonic Innervation: Immunohistochemistry Analysis

#### 2.7.1. Colonic Sections

Immunohistochemistry was performed on colonic paraffin-embedded sections that were 5 µm thick. Deparaffined slides were washed with phosphate-buffered saline (PBS) with 0.05% Tween 20 (Calbiochem, Darmstadt, Germany). Thereafter, sections were incubated for 10 min in 3% (*v*/*v*) hydrogen peroxide to inhibit endogenous peroxidase activity and blocked with horse serum for 30 min to minimize the nonspecific binding of the primary antibody. Sections were then incubated overnight at 4◦C with an anti-SP antibody (ab216412) (1:750, Abcam plc, Cambridge, UK) to perform a semi-quantitative analysis of the area reactive to SP in the myenteric ganglia of the colon. As a negative control, preparations were incubated without the primary antibody. The peroxidase-based detection kit ImmPress (Vector labs, Burlingame, CA, USA) was used as chromogen. Samples were coverslip-mounted with Eukitt mounting media (O. Kindler GmbH & Co, Freiburg, Germany) and studied under a Zeiss Axioskop 2 microscope equipped with the image analysis software package LAS X. Five photographs per animal were obtained and evaluated with the program Image J-Fiji (https://imagej.net/ accessed on 4 June 2024; open source under the General Public License) to measure the marked areas. A larger marked area indicates a higher concentration of SP cells in the myenteric ganglia of the colon.

#### 2.7.2. Whole-Mount Longitudinal Muscle–Myenteric Plexus Preparations 

To evaluate the effects of the regular intake of IC on the myenteric plexus, immunohistochemistry was employed on longitudinal muscle–myenteric plexus (LMMP) whole-mount samples [30,31]. Distal colon samples (2 cm long) were obtained, placed in Krebs solution, stretched, and pinned on a Sylgard^®^-coated dish (Farnell, Madrid). Mucosal and submucosal layers were manually removed by sharp dissection with fine forceps. The tissue was immersed in Zamboni’s fixative for 24–48 h at 4 °C and cleared with dimethyl sulfoxide (DMSO) (3 × 10 min) and PBS (3 × 10 min). Thereafter, the circular smooth muscle layer was removed, leaving only the longitudinal muscle layer with the myenteric plexus attached.

Preparations were stored at 4 °C in PBS with sodium azide (1%), until immunohistochemical processing was performed. For this, tissues were incubated 3 days (24 h at room temperature (RT), and the remaining 48 h at 4 °C) with a mixture of both, the pan-neuronal marker HuC/D (1:500; mouse biotin-conjugated, A-21271, from Thermo Fisher Scientific, Breda, The Netherlands) and sheep anti-nNOS (neuronal nitric oxide synthase, 1:500; Sigma-Aldrich, AB1529, from Merck, Darmstadt, Germany) antibodies. After washing with PBS (3 × 10 min), tissues were exposed for 24 h at RT to a mixture of streptavidin-AlexaFluor 488 (1:500; S11223, from Thermo Fisher), and donkey anti-sheep-RRX (1:500; 713-295-003, from Jackson, Cambridgeshire, UK). Finally, LMMP preparations were washed again with PBS (3 × 10 min), dehydrated in 50–70–100% buffered glycerol (10 min each) and mounted on slides. As a negative control to detect nonspecific labeling, tissue samples were also incubated in the absence of the primary antibody and the entire protocol was completed.

The preparations were observed under an inverted fluorescence Nikon Eclipse TE2000-U microscope (Nikon, Tokyo, Japan), equipped with a Nikon Moments camera with NIS-Elements software (Nikon, Tokyo, Japan, https://industry.nikon.com/en-gb/products/industrial-microscopy/industrial-microscopes/software/, accessed on 31 July 2024). Moreover, 5–7 whole-mount LMMP preparations per treatment were used. The analysis was performed blinded in 8–10 non-overlapping microphotographs (20× magnification) per preparation and marker using the program ImageJ-Fiji (https://imagej.net/ (accessed on 4 June 2024).

The general architecture of the myenteric plexus and the neuronal populations were analyzed. As a common criterion, ganglia were defined as agglomerates of three or more neurons. The following parameters were analyzed: the ganglionic area (expressed as percentage of the serosal surface area visualized), the ganglionic size, the number of neurons per ganglion, the total neuronal density (the number of neurons per serosal surface unit), the packing density (the number of intraganglionic neurons per ganglionic area), and the density of extraganglionic neurons (per serosal surface area). Also, the proportion of nNOS-positive neurons was calculated as the percentage of the neurons positive for HuC/D.

### 2.8. Drugs and Reagents

Atropine was acquired from Braun (Barcelona, Spain). ACh and the rest of the salts and reagents for Krebs solution preparation were obtained from Sigma-Aldrich (Darmstadt, Germany). To obtain the different concentrations of ACh, a stock of ACh 10^−2^ M diluted in distilled water was prepared. KCl was prepared in the laboratory, diluted in distilled water to a concentration of 4 M. Krebs solution was also prepared in the laboratory, fresh every day before starting the experiments, with the following composition (mM): 118 NaCl; 4.75 KCl; 1.2 MgSO_4_; 1.19 KH_2_PO_4_; 2.54 CaCl_2_; 25 NaHCO_3_; 11 glucose, pH 7.4 [28]. 

The reagents needed for the processing of histological sections and whole-mount preparations were as follows: 4% formaldehyde dissolved in PBS (formaldehyde 37%, Panreac Applichem, Castellar del Vallès, Spain, REF: 253572), paraffin (Histosec Pastilles, Merk Millipore, Madrid, Spain, REF: 1.11609.2504), hematoxylin (hematoxilyn 1-hidrate, Panreac Applichem, Castellar del Vallès, Spain, REF: 251344), eosin (eosin yellowfish, Panreac Applichem, Castellar del Vallès, Spain, REF: 25299), DMSO (DMSO 99,5% for synthesis, Panreac applychem, Castellar del Vallès, Spain, REF: 161954), sodium azide (sodium azide for analysis, Panreac Applychen, Castellar del Vallès Spain, REF: 122712) and glycerol (dissolved in PBS where appropriate) (glycerol for analysis, Panreac Applychen, Castellar del Vallès, Spain, REF: 131339). Zamboni’s fixative was prepared using 15% picric acid (Panreac Applychen, Castellar del Vallès, Spain, REF: A2520) and 20% paraformaldehyde (Panreac Applychen, Castellar del Vallès, Spain, REF: 141451) dissolved in PBS [30,31].

### 2.9. Data Evaluation 

Statistical evaluations were conducted using the GraphPad Prism 8 software (version 8). To analyze the functional data, all responses were normalized to the initial response to KCl. The chi-square test was used to analyze the distribution of females in each estrous cycle phases and the number of usable strips, both of which were shown as percentage. The remaining results were shown as the average ± standard error of the mean (SEM) and analyzed using one-way or two-way ANOVA. Bonferroni’s test was applied for multiple comparisons after ANOVA, while Tukey’s test was used for whole-mount LMMP preparations. Statistical significance was established for *p*-values below 0.05.

Different symbols were used in the graphs to highlight the statistically significant differences found in the study. First, # and $ illustrate sex-based differences (males vs. females: # without and $ with IC treatment). Second, * and + indicate differences due to the type of beverage administered (control vs. IC beverage; * for males; + for females). Finally, ° was used to stress the differences caused by atropine when compared to its corresponding nonatropine-treated group. 

## 3. Results

### 3.1. General Health

Throughout the three weeks of the *in vivo* study, the general well-being of the rats was assessed by tracking their body weight and solid and liquid intakes. All these data are compiled in the Appendix A. Additionally, at the end of the study, just before sacrifice, vaginal smears were performed in female rats to examine their distribution, as detailed in Appendix A.

### 3.2. Colon Wall Health: Histological Analysis

There were no statistically significant differences in the appearance of the colon wall between the groups (Figure 5A). Nonetheless, the thickness of the colonic muscle was notably reduced in females compared to males. Exposure to the IC beverage did not affect this measure when comparing IC animals to their respective controls (*p* > 0.05) (Figure 5B).

### 3.3. Colon Contractility: Organ Bath Study

Of the total number of strips obtained (four/muscle/rat), 71–83% were included in the study (because they reached the contractile response threshold and did not break throughout the study), with the following distribution: 71% LM and 79% CM for Males—Control, 79% LM and 79% CM for Males—IC, 75% LM and 75% CM for Females—Control and 83% LM and 83% CM for Females—IC, without statistical significance across groups (*p* > 0.05). 

#### 3.3.1. Responses to KCl Stimulation

After KCl administration, two different responses could be observed for both types of muscle, a phasic and a tonic one (Figure 4A), the tonic one (TA) with a lower amplitude but more prolonged than the phasic one (PA). 

Table 2 demonstrates that LM preparations exhibited a consistent response to initial KCl stimulation for both the phasic and tonic components, with no statistically significant differences observed between groups based on sex or IC exposure. Likewise, the response to the final KCl stimulation was homogeneous among groups. 

In contrast to the results obtained for LM strips, the phasic and tonic responses to the initial KCl in CM strips were significantly smaller in females than males in the control groups (*p* < 0.01 for the phasic response; *p* < 0.05 for the tonic response), as was the phasic response to the final KCl stimulation (*p* < 0.05). When comparing the groups exposed to IC, the initial response was similar to that in the control groups; thus, IC did not significantly modify the KCl-induced contractile response of the muscle strips (*p* > 0.05). Additionally, the final response to KCl was not statistically different from the initial one (*p* > 0.05). 

#### 3.3.2. Electrical Field Stimulation 

Electrical field stimulation (EFS) produced a two-phase response, with the first phase occurring within the initial 10 s of stimulation (A_1_) and a subsequent peak appearing immediately afterward (A_2_) (Figure 4B). To analyze the results, the amplitude of each response was normalized to the phasic response induced by the initial KCl administration. 

In the LM strips, the normalized amplitude of both A_1_ and A_2_ was frequency-dependent and similar across all groups, with no significant variations linked to sex or IC beverage consumption (Figure 6A,B). 

The results obtained for the CM strips were also frequency dependent. However, in A_1,_ this increase was relatively small and only happened at high frequencies (10–20 Hz). Thus, the maximum increase was around 50% of the response to KCl in the control male group and occurred with a stimulation of 20 Hz (Figure 6C). On the contrary, higher values were reached in A_2_ and these increased more progressively, starting at lower frequencies than in A_1_ (Figure 6D). Interestingly, EFS induced greater A_1_ and A_2_ amplitude values in females than males in this muscle, which reached values around 75% for A_1_ and 125% for A_2_ of the KCl response in females vs. 50% and 80% in males, respectively. These differences reached statistical significance for A_1_ at 20 Hz (Figure 6C) and for A_2_ at 0.1 and 2–20 Hz (Figure 6D). Exposure to the IC beverage did not modify the responses obtained in either sex for the A_1_ and A_2_ components of the CM response to EFS (Figure 6C,D).

In the presence of atropine (10^−6^ M), there was a significant reduction in the response to EFS in both muscles and both amplitudes (A_1_ and A_2_), although a non-muscarinic component remained after atropine in all responses, particularly at high EFS frequencies (Appendix A). To facilitate the comparison, the specific analysis of this component and the differences in it among experimental groups is shown in Figure 7. In LM, the non-muscarinic response of A_1_ showed a progressive increase across all groups from 5 Hz, depicted by overlapping lines, except in the case of IC males, where it showed a greater increase compared to their control group (around 40% of the KCl response compared with around 20% for the other groups), although these differences did not reach statistical significance (Figure 7A). Regarding A_2_ in LM, the non-muscarinic component was around 10% of the KCl response for females treated with IC at all frequencies and it increased to 20% for males at 20 Hz, presenting significant differences only by sex between control groups at this frequency (Figure 7B). Regarding EFS in CM, the non-muscarinic component of A_1_ increased at the highest frequencies (10 and 20 Hz) in all groups (up to 20–30% of the KCl response) in an overlapping manner, with no statistically significant differences among them (Figure 7C). In contrast, the atropine-resistant component of A_2_ for CM was minimal at 0.5 Hz, increased between 1 and 10 Hz and slightly decreased at 20 Hz in all groups. Interestingly, the non-muscarinic component of A_2_ in this muscle increased significantly in female compared with male groups and IC caused a further increase only in females, which at 10 Hz displayed responses of around 35% and 55% of the KCl response in control females and in females exposed to IC, respectively (Figure 7D). 

#### 3.3.3. Chemical Stimulation with Acetylcholine 

The contractions in longitudinal (LM) and circular (CM) muscle strips, induced by chemical stimulation with ACh were dependent on its concentration for both phasic (PA) and tonic (TA) responses. There were no significant differences observed between sexes or due to IC consumption (Figure 8). 

As expected, atropine completely abolished the muscle response to the maximum ACh concentration in all experimental groups (Figure 9). The non-muscarinic response remaining after atropine was less than 5% in all cases. 

### 3.4. Colonic Innervation: Immunohistochemistry Analysis 

#### 3.4.1. Substance P

SP expression was measured in the myenteric ganglia of the colon in histological sections. An analysis of SP immunoreactivity revealed no significant changes across the various study groups (Figure 10).

#### 3.4.2. Myenteric Plexus Study in Whole-Mount LMMP Preparations

The effects of IC intake on the structure of the myenteric plexus of male and female animals was analyzed using immunohistochemical techniques in whole-mount LMMP preparations that allow to evaluate a big number of myenteric ganglia and neurons from the same sample (Figure 11). 

As observed in Table 3, there were no statistically significant changes in the percentage of the area occupied by ganglia, the mean ganglionic size and the mean number of cells per ganglion. Likewise, there were no relevant modifications in the total neuronal density, neither in intraganglionic or extraganglionic density. Finally, the percentage of nNOS-positive neurons, mainly present in inhibitory neurons, was similar in male and female animals, regardless of the beverage, although in animals treated with IC, these values tended to be slightly higher than in their sex-matched controls, without the differences reaching statistical significance (Table 3).

## 4. Discussion

This study examined how regular IC consumption affects colonic motor function in both male and female rats. In general, sex was found to be the primary determinant of observed changes among experimental groups. In addition, our data intriguingly suggest that the regular intake of this beverage, which contains bioactive compounds such as polyphenols and caffeine, may modulate colonic motor function. Female rats demonstrated a greater IC consumption and a more pronounced contractile response of the circular muscle to electrical (neural) stimulation. However, no significant histological or immunohistochemical changes affecting the colonic innervation that could explain these functional results were detected.

### 4.1. Colon Wall Health

Our current histological study demonstrated a normal appearance of the colon wall in all groups of animals, without differences related to sex or IC exposure in the mucosa, submucosa and muscle layers, except for a relatively reduced colonic muscle thickness in females compared with males, possibly associated with their lower food intake and lower body weight, which are influenced by hormonal and genetic factors [32]. 

Furthermore, despite the fact that female rats exposed to IC consumed more liquid than their controls (Appendix A, ref. [21]), no structural changes were observed in their colon. nor did we find signs of inflammation in their colonic tissues after 4 weeks of IC consumption. Importantly, the distribution of female rats according to the phase of the estrous cycle did not display any statistically significant differences (Appendix A), discarding any specific influence of the hormonal status on the thickness of the colonic muscle between the two groups of females.

Importantly, we had previously observed that regular IC intake does not affect body weight gain (also seen here, Appendix A), behavioral parameters and general gastrointestinal transit in males or females [21]. Thus, our present results further suggest regular IC consumption is innocuous not only to the general health of the individuals but also to that of the colon wall.

### 4.2. Colon Contractility

The different stimuli that have been used in this study induce muscle contractions through different mechanisms: KCl causes a non-specific stimulation of all excitable elements of the colonic strips, ACh directly activates type 2 and 3 muscarinic receptors of the muscle, and EFS stimulates the myenteric neurons, causing the release of signaling molecules that bind to muscle receptors [33]. The responses to ACh and EFS were normalized to that of KCl for a clearer comparison among groups [28]. 

KCl stimulation caused a brief phasic contraction, then a tonic phase with lower amplitude and steady tone [29,34]. CM showed a lower response to KCl in the female groups, which is in line with the thinner colonic muscle observed in females when compared to males (Figure 9B). Additionally, the partial inhibition of Ca^2+^ flux through L-type channels in the smooth muscle cells by estrogen and progesterone has been observed in females [35,36], which could also explain the observed sex-dependent differences. Interestingly, IC beverage seemed to offset the decrease in response observed in the CM of females. This result supports the concept that regular IC intake may cause adaptive changes, at least in the colonic neuromuscular unit. 

Despite the greater liquid intake and relatively greater increase in KCl muscle response in IC-exposed females, there was no change in muscle thickness compared with their sex-matched controls, suggesting that the increased consumption of macronutrients (non-fiber carbohydrates, 47.48%; proteins, 6.25%; and lipids, 0.58% [13]), via increased IC consumption, did not cause any colonic smooth muscle hypertrophy in female rats. This could be due to a lack of impact of the macronutrients on smooth muscle tissue or because the food matrix’s micronutrients may counteract any hypertrophic effects. The impact of caffeine on colonic smooth muscle remains a subject of debate due to conflicting results from *in vitro* studies, indicating either smooth muscle relaxation or stimulation [37,38,39]. It is important to note that IC beverage has a lower caffeine content (32.64 mg per 240 mL) than coffee (80–100 mg per 240 mL) and a similar or lower caffeine content than tea (14–61 mg per 240 mL) and mate (30–90 mg per 240 mL [13]). This characteristic distinguishes the IC beverage and may have implications for the effect of its regular consumption on colonic motility and other aspects of gastrointestinal health. Therefore, the relatively increased colonic response to KCl in females after exposure to IC might be due to other bioactive compounds present in IC beverage, such as melanoidins. The regular intake of melanoidins could potentially strengthen muscle contractions by fiber effects that enhance peristalsis [16]. Although no hypertrophy was evident in this research, the muscle may adapt during ingestion due to fiber presence, with this adaptation being potentially observable *in vitro* in responses like those to KCl. To better understand these interactions, more specific research is needed on the long-term effects of dietary fiber on the contractility of gastrointestinal tissues. 

Unlike electrical stimulation, which induces a neuronal response, ACh administration directly activates muscarinic receptors of the smooth muscle tissue, also inducing muscle contraction [28,40,41]. Two distinct responses were identified: PA and TA. The PA response is primarily driven by calcium release from the sarcoplasmic reticulum, whereas the TA response is caused by the influx of extracellular Ca^2+^ [42]. The stimulation of muscle strips with ACh resulted in a dose-dependent increase in the strength of both contraction components in LM and CM. This effect was consistent across both sexes and equally sensitive to atropine, confirming that the responses were due to the activation of muscarinic receptors [43]. Furthermore, the intake of IC beverage caused no noticeable impact on muscle contraction in response to ACh compared with that obtained from control animals (drinking water). In a previous report using both male and female rats (without a specific evaluation of the possible sex-dependent effects) exposed (by gavage) for 3 days to regular or decaffeinated coffee, there was neither an effect on the *in vivo* colonic propulsion of a glass bead nor on the *in vitro* contractility of colonic full thickness strips obtained from these rats in response to ACh [44]. This was interpreted as a lack of genomic effects of the exposure to these beverages on the colonic neuro-musculature, and a similar conclusion could be drawn from our results obtained after a longer exposure to IC in drinking water. However, the contractile responses to electrical stimulation, which are mainly mediated by the neural release of neurotransmitters, was not evaluated in that study [44]. We did perform this type of analysis here. 

In general, in our study, all test groups exhibited a two-phase response to EFS, with both absolute and normalized amplitude increasing in a manner dependent on the stimulation frequency, reaching the maximum response at 10–20 Hz, in accordance with previous studies [28,45,46]. In this biphasic response, one contraction is observed during the electrical stimulation (A_1_) and a second one occurs once the stimulation has ended (A_2_), often termed an “off-response”; this biphasic response has been previously described in several studies both in animals [28,47] and humans [29,48,49]. Finally, there is a relaxation mediated by the release of nitrergic inhibitors, such as NO [45], which prevents A_2_ from being prolonged over time in the form of a tonic response. This agrees with the mean duration of A_2_ (Figure 4B), which was 10–35 s, typical of a phasic colonic response. In the different experimental groups, atropine inhibited both A_1_ and A_2_ responses in both muscles, although, compared with the response to ACh, in this case, a significant non-muscarinic component remained, which was well above 10% of the KCl response, particularly at the highest frequencies tested (10–20 Hz). According to previous studies, A_1_ is mainly mediated by the release of ACh from myenteric neurons and the activation of muscarinic receptors [29], although purinergic activation may also occur at low frequencies [45]. Similarly, according to Aulí (2008) and Broad (2012) [45,49], the off-response (A_2_) is inhibited by the joint action of atropine and neurokinin receptor antagonists; thus, it is due to the release of both ACh and SP [45] from the myenteric neurons in response to EFS. SP, through the NK1 receptor, can independently trigger contractions in gut smooth muscle, thereby compensating for the absence of a cholinergic component [50]. 

Despite the obvious similarities in the frequency-dependent contractility pattern of colonic muscle strips in all experimental groups, there were some relevant quantitative differences in our study. In particular, the amplitude of the CM contractions was greater in females than that of control males, especially that of the A_2_ response, at 0.1 and 2–20 Hz. These results are consistent with those obtained in a human colon research study by Drimousis (2020) [51]. Since no sex-dependent change was found in the response to ACh, the increase in the response of CM strips to EFS in females compared to males could be due to differences in the myenteric plexus or variations in neurotransmission. 

Although exposure to IC did not significantly impact the contractility of colonic strips induced by EFS in either the CM or LM layers, atropine was less effective in inhibiting CM A_2_ responses to elevated-frequency EFS in IC-treated females, suggesting a greater involvement of non-muscarinic pathways in these responses. Importantly, there were no substantial variations in the estrous cycle phase between the two female groups at the time of the organ bath studies, indicating that the functional differences observed could be solely due to the impact of regular exposure to IC and its individual components on the colonic intrinsic innervation and not to the hormonal status of the animals at sacrifice. 

Thus, we next evaluated the possible impact of the differences in sex and beverage exposure on the colonic myenteric plexus.

### 4.3. Colonic Innervation

Although females had smaller myenteric ganglia with fewer intra-ganglionic neurons than males, these differences did not reach statistical significance. Therefore, our functional results do not seem to be attributable to general sex-dependent changes in the myenteric innervation of the colon. However, they could be due to an increase in the excitatory and/or a decrease in the inhibitory components involved in neurotransmission in the myenteric plexus.

In the myenteric plexus, the main inhibitory neurotransmitter is NO [4,52,53]. Therefore, we evaluated the proportion of nNOS-immunoreactive myenteric neurons in whole-mount LMMP preparations but did not find any significant differences between sexes in this parameter (Table 3). Thus, our results suggest that the increased contractility of CM in females is not due to a decrease in the population of inhibitory myenteric neurons. Moreover, the lack of statistically significant changes in the myenteric structure and the proportion of nNOS inhibitory neurons in either male or female rats due to IC exposure suggests this beverage did not exert potent effects on myenteric nerve fiber distribution and nNOS activity, although more subtle modulatory effects cannot be discarded at this stage. 

After ACh, the main excitatory myenteric neurotransmitter is SP, which activates NK receptors in the smooth muscle and is released in response to high frequency EFS [45,51]. Consequently, we quantified the myenteric area immunoreactive to SP in histological sections through immunohistochemistry. Under our experimental conditions, we did not observe any significant difference in SP expression in the myenteric plexus between control males and females, which is in contrast with a previous study [51]. Furthermore, there was no significant rise in myenteric SP expression in rats of either sex given IC, compared to their control groups. However, despite a similar amount of SP seems to be present in the myenteric plexus, it might be more efficient to contract the colonic muscle in females, particularly those drinking IC, through more abundant or more efficient tachykinergic receptors. Alternatively, other excitatory neurotransmitters could be more involved in the non-muscarinic component of the colonic contraction in females after IC exposure. 

Whatever the case may be, the presence in IC of melanoidins, polyphenols, and caffeine, compounds known for their strong antioxidant properties, may have contributed to our functional findings by protecting neuronal cells in the colon and enhancing the function and resilience of the neural pathways [14]. Additional studies are required to elucidate the specific mechanisms involved.

### 4.4. Significance, Strengths and Limitations

Products derived from coffee cascara, traditionally used in Central America and Africa, were approved as novel foods by the EFSA in 2022, highlighting their safety and potential value for human health. Our previous results using IC beverage in an *in vivo* study were in accordance with this concept [21]. However, these products might be able to exert certain effects on specific tissues. These effects may only be unmasked when evaluated *in vitro*.

Using animal models allows for preliminary data collection in a highly controlled environment, enabling the evaluation of new products’ effects, a better understanding of the mechanisms of action, and the identification of potential side effects. Our study was conducted using colon samples from healthy rats that had consumed an IC beverage on a regular basis. Although organ bath studies can also be carried out using human colon samples [29,45], due to ethical and practical restrictions, these cannot be obtained from healthy individuals (i.e., samples are obtained from colorectal cancer patients undergoing surgery). Furthermore, studies using both animal and human samples are scarce and generally test the direct effects of the food extracts or ingredients when added to the organ bath [29], whereas the studies evaluating the effects of their acute or chronic consumption are practically restricted to experimental animals [44]. These are even more scarce but essential to delineate the plastic changes that certain foods may cause in the long run in the functioning of the organs in healthy individuals. Thus, despite the fact that interspecies variations limit direct extrapolation to humans, our results remain valuable for establishing a preliminary basis.

In our study, the rats were grouped in their cages and IC was ingested on an “ad libitum” manner, thus it was not possible to accurately measuring the exact amount of IC consumed each day by each rat. To increase dosing consistency, multiple strategies can be implemented. Administering IC via gavage would allow a “pharmacological” approach since the dose given to each animal would be adjusted by its weight [44]. Individually tracking intake (using metabolic cages) could also be helpful to more accurately determine the exact amount of IC consumed by each rat. However, both repeated gavage and individual caging cause significant stress to rodents, impacting the results [54,55,56,57]. Stress is an important factor affecting the functioning of the brain–gut axis in general [58] and gastrointestinal motor function in particular [59]. Importantly, compared to the animals consuming plain water, the average liquid intake was not reduced in either males or females exposed to IC, which could have led to some degree of dehydration, confirming IC was well accepted by the animals at the dose tested [21]. Although modern approaches using video recording and artificial intelligence might be helpful in more precisely measuring individual IC consumption under our non-invasive conditions, the design used in this study resembled natural feeding, minimizing stress in the animals and enhancing the results’ validity.

Previous *in vivo* studies from our laboratory have described important differences between sexes in the motor and sensitive functions of the rat gastrointestinal tract [24,60]. The present study shows that subtle plastic changes occur in a sex-dependent manner at the tissue level after a relatively short exposure to IC, even in the absence of *in vivo* differences in gastrointestinal transit between control and IC-exposed male and female animals [21]. The absence of specific controls for components such as chlorogenic acid and caffeine limits the understanding of their individual effects but does not invalidate the general findings on IC as a food matrix. Furthermore, our results may be particularly relevant in the context of IBS and other functional gastrointestinal disorders (FGIDs), which are highly prevalent in females and highly sensitive to certain food components that may either trigger or alleviate them [61]. More research is warranted to determine whether IC-induced changes may protect against the development of FGID symptoms or facilitate them. 

Overall, the risk/benefit balance of the experimental design supports its utility in providing valuable preliminary data while acknowledging areas for improvement in future research.

## 5. Conclusions

Our findings indicate that the regular consumption of IC induces subtle adjustments in colonic muscular contractility, possibly linked to modifications in the release of neurotransmitters or their activity. The enhanced neural reactivity observed in females exposed to IC provides compelling evidence of sex as a critical factor in the effects of foods regularly consumed. Consequently, this study not only adds to the growing field of gender-specific nutrition but also highlights the potential of IC’s bioactive compounds as promising candidates for targeted interventions in colonic health disorders. Of note, compounds composing IC possess antioxidant and anti-inflammatory properties with the potential to preserve neural and intestinal health. 

These findings underscore the significance of investigating the effects of bioactive compounds, not only when added to isolated systems but more importantly, when regularly consumed *in vivo* (included in food matrices) and support the value of animal models in advancing our understanding of complex health issues. 

## Figures and Tables

**Figure 1 foods-13-02474-f001:**
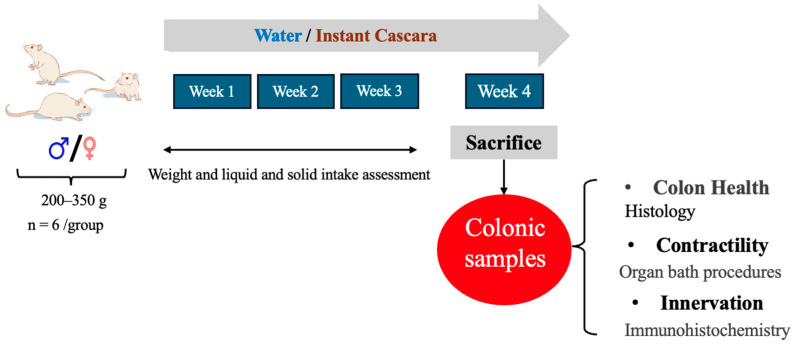
Experimental procedure. Male and female rats were given either water (control group) or IC beverage for a duration of 4 weeks. During the fourth week, animals were sacrificed to analyze the health of the colon wall using histological methods, colonic muscle strip contractility employing organ bath procedures, and its innervation using immunohistochemistry techniques.

**Figure 2 foods-13-02474-f002:**
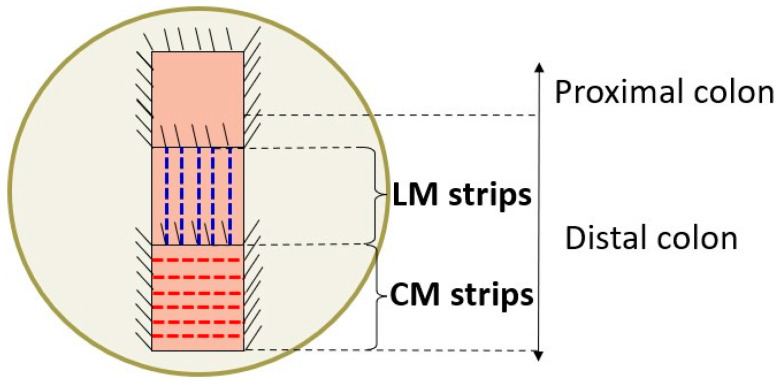
Schematic representation of how the longitudinal and circular muscle strips were obtained. Once the colonic segment was removed from the rat, it was pinned on a Petri dish covered with Sylgard^®^ and filled with Krebs solution. A longitudinal cut was performed through the mesenteric border. Once stretched and pinned on the dish surface, the mucosa and submucosa layers were removed, and the circular (CM) and longitudinal (LM) muscle strips were obtained by cutting perpendicular or parallel to the longitudinal axis of the colon, respectively.

**Figure 3 foods-13-02474-f003:**
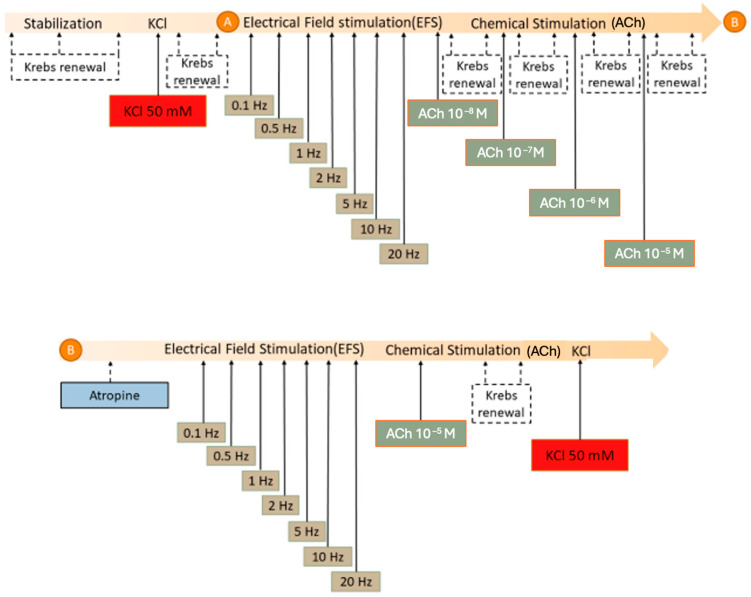
Experimental protocol of the organ bath experiments. Longitudinal and circular muscle strips were suspended in organ bath cups. Upper panel: After an initial 60 min stabilization period with 3 Krebs renewals, potassium chloride (KCl) was added at 50 mM to study the contractility of the strips. (**A**) After 2 Krebs renewals, the strips were electrically stimulated (EFS) at increasing frequencies (0.1–20 Hz) and posteriorly with acetylcholine (ACh) at increasing concentrations (10^−8^–10^−5^ M), Krebs was renewed two times before the next concentration was added. Lower panel (**B**): The same electrical and chemical stimulations were repeated in the presence of atropine (10^−6^ M), only this time just with ACh 10^−5^ M and without Krebs renewals. Finally, Krebs was renewed twice, and KCl (50 mM) was added to the organ bath.

**Figure 4 foods-13-02474-f004:**
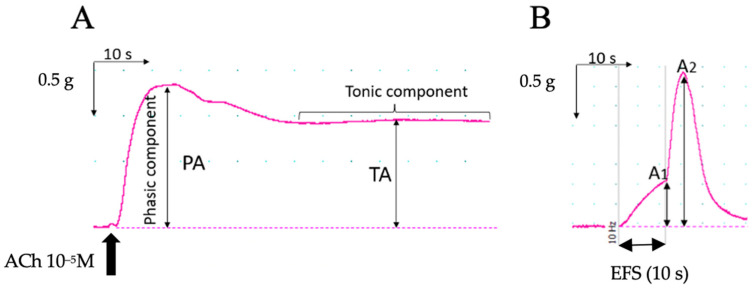
Representative traces of colonic smooth muscle contractile responses and parameters measured. (**A**) Measurement of the amplitude of the phasic (PA) and tonic (TA) components of the contractions induced by ACh (and KCl). TA occurs after PA, as a plateau, generally below the value of PA. (**B**) Measurement of Amplitude 1 (A_1_) and Amplitude 2 (A_2_) of the contraction induced during and after electrical stimulation (EFS), respectively. Thick vertical arrow in A and thin double-head arrow in B represent stimulus (administration of ACh in (**A**), EFS duration in (**B**)).

**Figure 5 foods-13-02474-f005:**
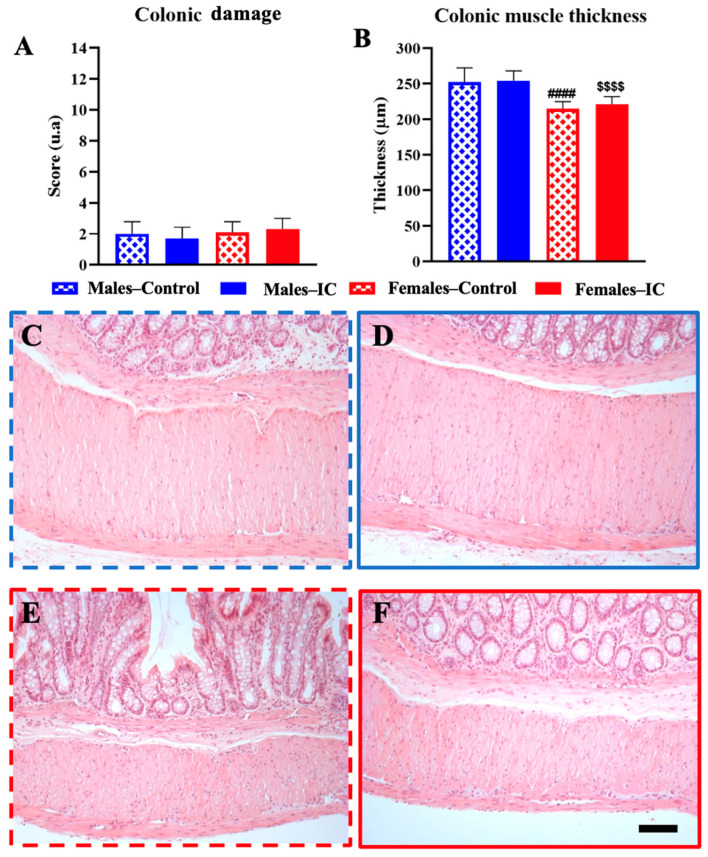
The impact of Instant Cascara (IC) beverage regarding the microscopic features of the colon in male and female rats. Colonic damage (**A**) was evaluated by examining ten randomly selected fields per section at 40× magnification, using three distinct sections of colon tissue for each animal. In addition, the width of the muscle layer was measured (**B**). The images (**C**–**F**) show the muscle layer of the colon. During the fourth week of IC beverage administration, tissue samples were collected from animals across four experimental groups: Males—Control, Males—IC, Females—Control, and Females—IC. Each group consisted of six animals. The results are presented as mean ± SEM (standard error of the mean). Significant differences related to sex were observed, with *p* < 0.0001 indicated by #### for comparisons between Females—Control and Males—Control, and $$$$ for comparisons between Females—IC and Males—IC. Statistical analysis was conducted using one-way ANOVA with Bonferroni’s post hoc test. Bar: 100 μm.

**Figure 6 foods-13-02474-f006:**
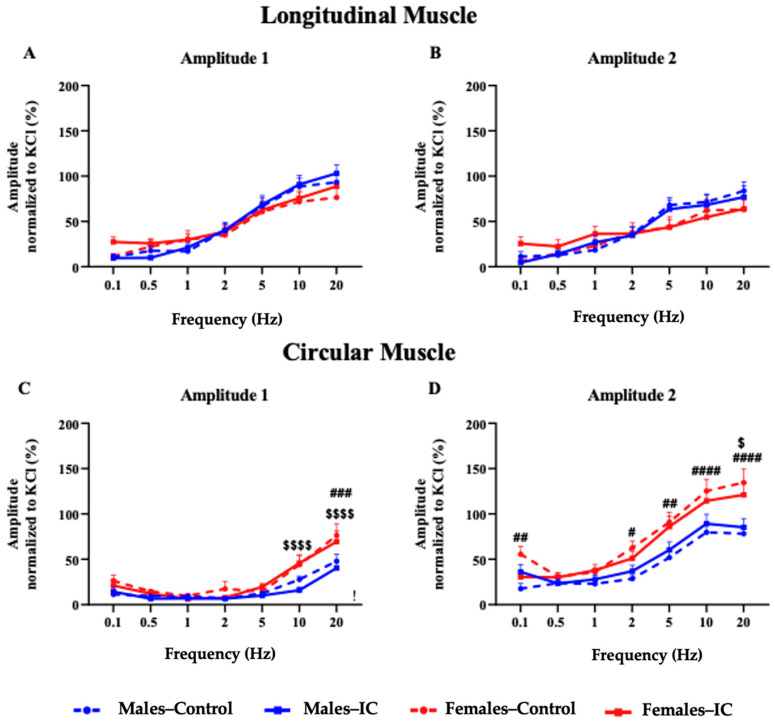
Electrical field stimulation (EFS) was applied to longitudinal (LM) and circular (CM) muscle strips using 10 s pulse trains (0.3 ms) at frequencies of 0.1, 0.5, 1, 2, 5, 10, and 20 Hz. The following metrics were recorded: (**A**) maximum amplitude observed during stimulation (Amplitude 1) in LM; (**B**) maximum amplitude detected after stimulation (Amplitude 2) in LM; (**C**) Amplitude 1 in CM; (**D**) Amplitude 2 in CM. Data are reported as mean ± SEM, with each group consisting of 5–6 rats and 17–20 muscle strips for both LM and CM. Statistically significant differences related to sex are indicated by the following: # *p* < 0.05; ## *p* < 0.01; ### *p* < 0.001; #### *p* < 0.0001 for Male—Control vs. Female—Control comparisons; $ *p* < 0.05; $$$$ *p* < 0.0001 for Male—IC vs. Female—IC comparisons. Statistical significance was determined using two-way ANOVA followed by Bonferroni’s post hoc test.

**Figure 7 foods-13-02474-f007:**
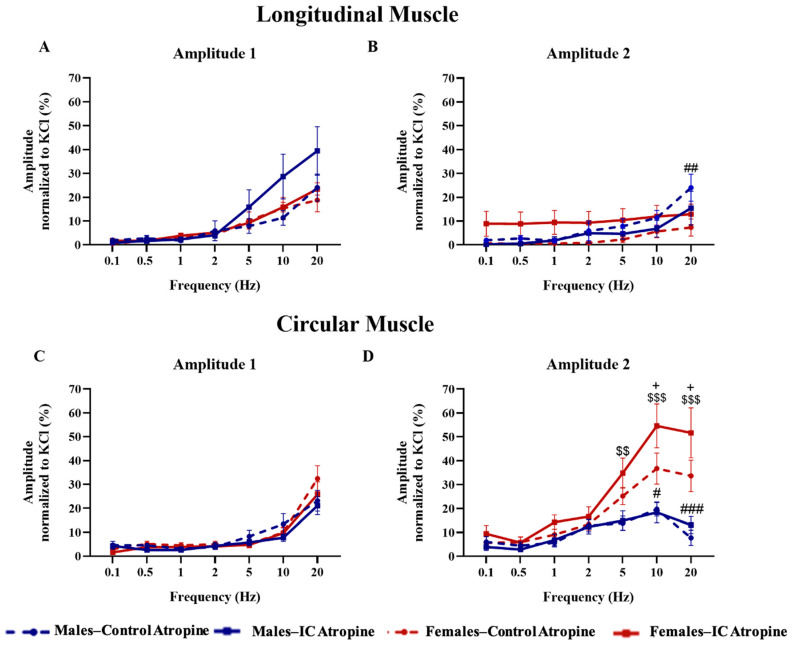
The effects of electrical field stimulation (EFS) on longitudinal (LM) and circular (CM) muscle strips were evaluated under non-muscarinic conditions after atropine (10^−6^ M). EFS comprised 10 s pulse sequences (0.3 ms, 100 V) across frequencies of 0.1, 0.5, 1, 2, 5, 10, and 20 Hz. The following were recorded: (**A**) peak amplitude during stimulation (Amplitude 1) in LM; (**B**) peak amplitude post stimulation (Amplitude 2) in LM; (**C**) Amplitude 1 in CM; (**D**) Amplitude 2 in CM. Data are reported as mean ± SEM for each experimental condition, with 5–6 rats and 17–20 muscle strips per condition. Sex-related significant differences are indicated as follows: # *p* < 0.05; ## *p* < 0.01; ### *p* < 0.001 comparing Male—Control to Female—Control; $$ *p* < 0.01; $$$ *p* < 0.001 comparing Male—IC to Female—IC. Significant beverage-related differences are shown by + *p* < 0.05 for Females—IC versus Females—Control. Analysis was performed using two-way ANOVA with Bonferroni’s post hoc test.

**Figure 8 foods-13-02474-f008:**
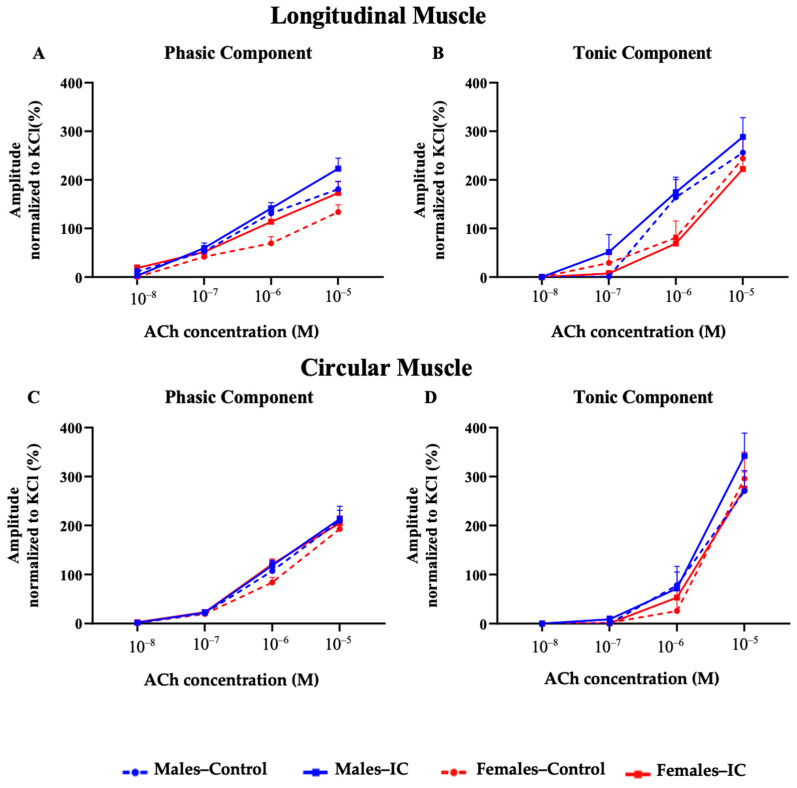
The contractile responses of longitudinal (LM) and circular (CM) muscle strips to acetylcholine (ACh) stimulation were examined by exposing the samples to a range of ACh concentrations (10^−8^ to 10^−5^ M). The data illustrate the phasic (**A**) and tonic (**B**) responses in LM and the phasic (**C**) and tonic (**D**) responses in CM. Results are presented as mean ± SEM for each group, with six rats per group and 17–20 strips per group for both LM and CM. The analysis was conducted using two-way analysis of variance (ANOVA) with subsequent Bonferroni’s correction for multiple comparisons, and no significant differences were observed (*p* > 0.05).

**Figure 9 foods-13-02474-f009:**
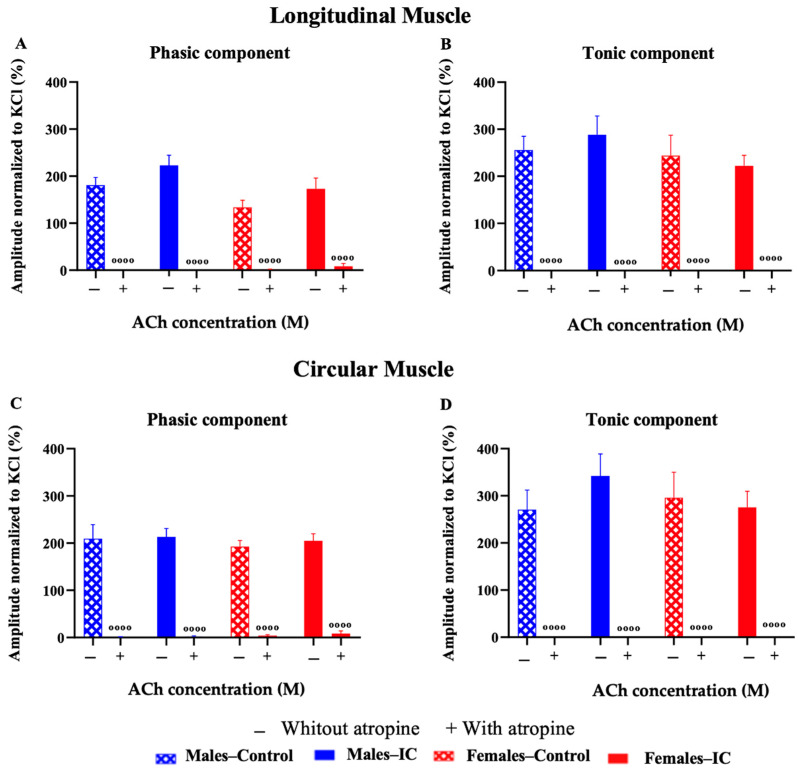
Contractile responses of longitudinal (LM) and circular (CM) muscle strips to acetylcholine (ACh, 10^−5^ M) were evaluated both before (−) and after (+) atropine (10^−6^ M) administration. This setup was used to determine the role of muscarinic receptors in the ACh-induced contractions. The amplitude of the contractile responses, including both the phasic (PA) and tonic (TA) components, were compared for LM and CM under these conditions, specifically: (**A**) phasic response (PA) in LM; (**B**) tonic response (TA) in LM; (**C**) phasic response (PA) in CM; (**D**) tonic response (TA) in CM. Data were expressed as mean ± SEM for each group, with six rats and 17–20 strips per group for both LM and CM. Significant differences between atropine-treated and non-treated groups are indicated by ᵒᵒᵒᵒ *p* < 0.0001. Statistical evaluation was performed using two-way analysis of variance (ANOVA) followed by Bonferroni’s post hoc test.

**Figure 10 foods-13-02474-f010:**
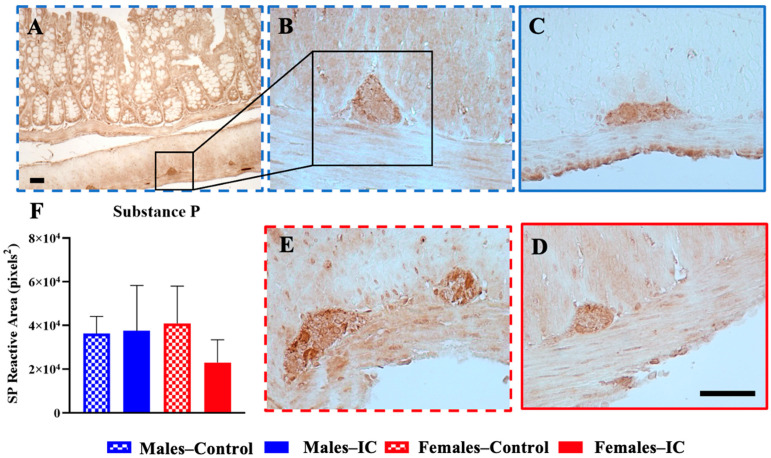
Impact of Instant Cascara (IC) beverage on the immunoreactivity of substance P (SP) within the colonic myenteric ganglia of male and female rats. Image (**A**) shows the myenteric ganglia of the colon, where SP is found. Images (**B**–**E**) show the myenteric ganglia immunoreactive to SP in the different experimental groups. The area immunoreactive to SP in the myenteric ganglia of the colon sections was quantified with the program Image J-Fiji (**F**). After a four-week period of administering the IC beverage, animals were sacrificed to evaluate various parameters across four experimental groups classified by sex and beverage type: Males—Control, Males—IC, Females—Control, and Females—IC. Results are expressed as mean ± SEM (standard error of the mean) for each group, with six animals per group. Statistical analysis was performed using a one-way analysis of variance (ANOVA), followed by Bonferroni’s post hoc test, with no significant differences found (*p* > 0.05). Bar: 50 μm.

**Figure 11 foods-13-02474-f011:**
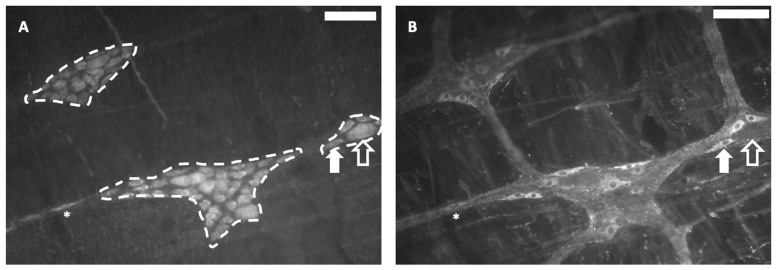
Whole-mount longitudinal muscle–myenteric plexus (LMMP) preparations were processed with the pan-neuronal marker HuC/D to detect all neurons (**A**) and neurons expressing neuronal nitric oxide synthase (nNOS), which is mainly present in myenteric neurons involved in inhibitory motor circuits (**B**). Using the program ImageJ-Fiji, the area occupied by the ganglia was measured (dashed lines (**A**)). Neurons immunoreactive for each of the mentioned markers were counted (**A**,**B**). Asterisk: extraganglionic neuron. White arrow: neuron positive for HuC/D and nNOS. Empty arrow: neuron positive for HuC/D and negative for nNOS. Scale bar: 100 µm.

**Table 1 foods-13-02474-t001:** Composition of Instant Cascara (IC) beverage.

Components	Instant Cascara Beverage (10 mg/mL)
Carbohydrates (mg/mL)	4.748
Total fiber (mg/mL)	1.832
Lipids (mg/mL)	0.058
Protein (mg/mL)	0.625
Magnesium (mg/mL)	2.084 × 10^−3^
Sodium (mg/mL)	26.658 × 10^−3^
Potassium (mg/mL)	228.4 × 10^−3^
Calcium (mg/mL)	5.478 × 10^−3^
Caffeine (mg/mL)	0.139
Phenolic content (mg eq. Cga/mL)	0.89
Chlorogenic acids (mg/mL)	1.07–1.26
Melanoidins (mg/mL)	1.5

The concentration of key nutritional and chemical constituents of IC beverage, including macronutrients, minerals, and bioactive compounds, is detailed. Notably, the phenolic content is measured as 0.89 mg equivalent to caffeic acid per milliliter (mg eq. Cga/mL), providing an indication of its antioxidant potential [13,23].

**Table 2 foods-13-02474-t002:** Muscle strip contractile responses to KCl stimulation.

		Males—Control		Males—IC		Females—Control		Females—IC	
		Initial (g)	Final (g)	Initial (g)	Final (g)	Initial (g)	Final (g)	Initial (g)	Final (g)
LM	PA	0.40 ± 0.07	0.35 ± 0.06	0.38 ± 0.06	0.31 ± 0.05	0.41 ± 0.05	0.33 ± 0.05	0.44 ± 0.05	0.36 ± 0.05
	TA	0.24 ± 0.04	0.21 ± 0.05	0.27 ± 0.05	0.18 ± 0.03	0.18 ± 0.03	0.19 ± 0.05	0.22 ± 0.03	0.19 ± 0.03
CM	PA	1.73 ± 0.21	1.53 ± 0.19	1.47 ± 0.18	1.48 ± 0.18	0.96 ± 0.13 **^##^**	0.96 ± 0.10 **^#^**	1.14 ± 0.09	1.31 ± 0.08
	TA	0.75 ± 0.11	0.70 ± 0.12	0.80 ± 0.11	0.63 ± 0.11	0.55 ± 0.09 **^#^**	0.37 ± 0.05	0.74 ± 0.09	0.54 ± 0.06

Contractile responses to 50 mM KCl were measured at both the initial and final stages of the experiment in colonic smooth muscle strips oriented longitudinally (LM) and circularly (CM). Moreover, two different responses could be observed for both types of strips, a phasic (PA) and a tonic one (TA). Statistical analysis was performed row-wise, corresponding to each parameter of each strip type. The study involved four groups based on sex and the type of beverage administered (IC or water): Males—Control, Males—IC, Females—Control, and Females—IC. The results are shown as mean ± SEM (standard error of the mean), with each group containing 5–6 animals. Notable sex-related differences were observed, with significance levels of # *p* < 0.05 and ## *p* < 0.01 for comparisons between Females—Control and Males—Control. Data were analyzed using one-way ANOVA with Bonferroni’s post hoc test for multiple comparisons.

**Table 3 foods-13-02474-t003:** Structural parameters and neuronal counts in colonic whole-mount LMMP preparations.

Antibody	Parameter (Unit)	Males—Control	Males—IC	Females—Control	Females—IC
HuC/D	Ganglionic area (%)	10.90 ± 1.21	10.59 ± 1.28	10.95 ± 0.59	11.05 ± 1.28
	Ganglionic size (μm^2^)	25,716 ± 2690	22,345 ± 2750	20,332 ± 3277	25,126 ± 3199
	Neurons/ganglion	54.06 ± 4.933	47 ± 4.77	41.91 ± 9.12	49.77 ± 6.431
	Total neuronal density (neurons/mm^2^)	231.40 ±19.46	212 ± 17.64	217.40 ± 23.49	219.90 ± 28.79
	Packing density (intraganglionic neurons/ganglionic area)	2156 ± 226.1	2147 ± 134.7	2000 ± 143.1	1976 ± 45.15
	Extraganglionic neuronal density (extraganglionic neurons/serosal area)	3.75 ± 0.78	4.28 ± 0.89	4.81 ± 0.35	3.03 ± 1.026
nNOS	Inhibitory neurons (%)	18.67 ± 2.24	22.13 ± 2.36	18.93 ± 2.02	22.25 ± 4.02

Results are presented as mean ± SEM based on 5–6 animals per group. Statistical comparisons were made using one-way analysis of variance (ANOVA) with subsequent Tukey’s post hoc test, showing no significant differences (*p* > 0.05). Abbreviations: nNOS, neuronal nitric oxide synthase.

## Data Availability

The original contributions presented in the study are included in the article and Appendix A, further inquiries can be directed to the corresponding authors.

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
