# Peer review of "Ex Vivo Study of Colon Health, Contractility and Innervation in Male and Female Rats after Regular Exposure to Instant Cascara Beverage"

_foods, 2024, doi:10.3390/foods13162474_

Round 1

Reviewer 1 Report

Comments and Suggestions for Authors

In the manuscript (ID: foods-3117616), authors researched the effects of regular exposure to Instant Cascara beverage on colonic health, contractility and innervation in male and female rats. In general, the content meets the requirements of Foods. Therefore, I think the manuscript can be accepted and published in Foods after a minor revision. To ensure the publication readiness of the manuscript, the following issue needs to be addressed:

(1) Abstract

--Line 38-39: Should be “Instant Cascara (IC)” rather “Instant Cascara (IC)”.

--Line 38-39: Should be “substance P (SP) and neuronal nitric oxide synthase (nNOS)” rather “SP (substance P) and nNOS (neuronal nitric oxide synthase)”.

--Line 42: Should be “which had” rather “which had had”.

(2) Introduction: The introduction provides an adequate background on the topic.

--Line 58-59: Should be “ensuring the digestive system functions effectively” rather “to ensure the digestive system functions effectively”.

(3) Materials and Methods

--Materials and reagents are omitted in this part. Suggest authors to add them. In addition, please check carefully and add information on relevant Materials and reagents. It is of utmost importance this is clarified and more detailed to allow replication.

--Line 287: Should be “longitudinal muscle-myenteric plexus (LMMP)” rather “LMMP”.

(8) Results and discussion

--The composition of Instant Cascara (IC) has significant influence on the experimental results. If possible, it is recommended that the author provide information on the composition of the IC.

--Line 350: “p” should be in italics. In addition, there are similar errors in other parts of the manuscript, and authors are advised to check the whole manuscript carefully and correct these minor errors.

--Line 400: Should be “KCl” rather “potassium chloride (KCl)”.

--Line 561: INSTANT CASCARA (IC) or Instant Cascara (IC)? It is suggested that the author unify its writing in this manuscript.

--Line 677: “in vivo” should be in italics. In addition, there are similar errors in other parts of the manuscript, and authors are advised to check the whole manuscript carefully and correct these minor errors.

--Line 735-736: Should be “the increased contractility of CM in females is” rather “the increased contractility of CM in females are”.

Comments on the Quality of English Language

Minor editing of English language required

Reviewer 2 Report

Comments and Suggestions for Authors

The authors here presented interesting research, considering the importance of colonic health and bioactive ingredients.

However, few things should be addressed prior the acceptance for publication.

The authors should explain how the tested concentration was chosen

Also, can the results be related to the human health and how

Finally, the strengths and the limitations of the study/research should be outlined in the discussion section. 

Reviewer 3 Report

Comments and Suggestions for Authors

The manuscript Ex vivo study of colon health, contractility, and innervation in 2 male and female rats after regular exposure to Instant Cascara 3 beverage presented a lot of results, and a lot of work was done. 

Introduction: Well written, it situates the reader on the subject.

Material and methods: Formalin, isn't it 4% concentrated, since formaldehyde comes in a concentration of 40% and not 100%?

Results: Well described, statistical analyses performed.

Discussion: when we analyzed the results, and there was no statistical difference, we cannot say that there was a tendency to improve or increase, there was no statistical difference, there was no change in relation to the negative control.

I suggest that it be clear in the results that the animals treated with the drink for the given time, at the appropriate concentration, showed these results.

For future studies, I suggest that animals receive the drink by gavage, once a day, in a concentration determined by the weight of each animal, to ensure that all animals receive the same dose.
